# The Expression of ERK1/2 in Female Yak (*Bos grunniens*) Reproductive Organs

**DOI:** 10.3390/ani10020334

**Published:** 2020-02-20

**Authors:** Jiangfeng Fan, Xiaohong Han, Honghong He, Yuzhu Luo, Sijiu Yu, Yan Cui, Gengquan Xu, Libin Wang, Yangyang Pan

**Affiliations:** 1College of Veterinary Medicine, Gansu Agricultural University, Lanzhou 730070, China; hanxiaohong11@126.com (X.H.); honghong3h@126.com (H.H.);; 2College of Animal Science and Technology, Gansu Agricultural University, Lanzhou 730070, China; 3Technology and Research Center of Gansu Province for Embryonic Engineering of Bovine and Sheep & Goat, Lanzhou 730070, China

**Keywords:** yak, ERK1/2, ovary, oviduct, uterus

## Abstract

**Simple Summary:**

Extracellular signal-regulated kinases1/2 (ERK1/2) plays a significant role in regulating the reproductive processes of mammals. The goal of our research is to investigate the expression and distribution of ERK1/2 in the main reproductive organs of the yak during different stages. Using immunohistochemistry, western blot, and relative quantitative real-time polymerase chain reaction techniques, we found that the expression of ERK1 and ERK2 proteins and their mRNA in the yak’s ovary, oviduct, and uterus varies with the stage of the reproductive cycle. The variation character of ERK1 and ERK 2 expression in the yak’s main reproductive organs during different stages implies that they play an important role in regulating the reproductive functions under different physiological statuses.

**Abstract:**

The main reproductive organs undergo different histological appearances and physiological processes under different reproductive statuses. The variation of these organs depends on a delicate regulation of cell proliferation, differentiation, and apoptosis. Extracellular signal-regulated kinases1/2 (ERK1/2) are members of the mitogen-activated protein kinase (MAPK) super family. They have important roles in regulating various biological processes of different cells, tissues, and organ types. Activated ERK1/2 generally promotes cell survival, but under certain conditions, ERK1/2 also have the function of inducing apoptosis. It is widely believed that ERK1/2 play a significant role in regulating the reproductive processes of mammals. The goal of our research is to investigate the expression and distribution of ERK1/2 in the yak’s main reproductive organs during different stages. In the present study, samples of the ovary, oviduct, and uterus of 15 adult female yak were collected and used in the experiment. The ERK1/2 proteins, localization, and quantitative expression of their mRNA were investigated using immunohistochemistry (IHC), western blot (WB) and relative quantitative real-time polymerase chain reaction (RT-PCR). The results indicated that ERK1/2 proteins and their mRNA were highly expressed in the ovary of the luteal phase and gestation period, in the oviduct of the luteal phase, and in the uterus of the luteal phase and gestation period. Immunohistochemical analysis revealed a strong distribution of ERK1/2 proteins in follicular granulosa cells, granular luteal cells, villous epithelial cells of the oviduct, endometrial glandular epithelium, and luminal epithelium. These results demonstrated that the expression of ERK1 and ERK2 proteins and their mRNA in the yak’s ovary, oviduct, and uterus varies with the stage of the reproductive cycle. The variation character of ERK1 and ERK 2 expression in the yak’s main reproductive organs during different stages implies that they play an important role in regulating the reproductive function under different physiological statuses.

## 1. Introduction

Extracellular signal-regulated kinases (ERKs) are an important subfamily of mitogen-activated protein kinases (MAPKs), which regulate various cellular activities and physiological processes. Originally, the ERK gene was isolated from the expression library of human gastric cancer in 1993 [1,2], and it was recognized as a genomic DNA, encoding a sequence of the receptor protein-tyrosine kinase. The ERK gene is evolutionarily conserved, and is found in all eukaryotes, from yeast to humans [3]. Activated ERK mediates extracellular signals (transferring from cell membrane receptors to cytoplasm and nuclear effectors) and regulates some specific gene expression by phosphorylating transcription factors; thus, participates in the regulation of cell growth, development, differentiation, and proliferation. By now, it is well known that the classical ERK cascade consists of Rafs (MAP3K), mitogen-activated protein kinase 1/2 (MEK1/2), extracellular signal-regulated kinases1/2 (ERK1/2), and several MAPK-activating protein kinases (MAPKAPKs) [4]. ERK1/2, also known as mitogen-activated protein kinases 3 and 1 (MAPK3/1), is one of the major MAPK cascades. Thus, MEK1/2, MAP3Ks, and other upstream signals can persistently, or transiently, activate ERK1/2. Activated ERK1/2 generally promote cell survival, but under certain conditions, ERK1/2 also have the function of inducing apoptosis [5].Many different stimuli, including growth factors [6], radiation [7,8], osmotic stress [9,10], Fas ligand [11], nitric oxide [12], and hydrogen peroxide [13] activate the ERK1/2 pathway. Remarkably, some researchers have indicated that activated ERK1/2 can inhibit apoptosis induced by hypoxia condition [14,15,16,17]. Thus, it is widely believed that ERK1/2 play an important role in mammalian organ development, incorporating with cellular proliferation, differentiation, migration, fate determination, growth, and apoptosis [18]. There are a number of materials reporting the expression of ERK1 and ERK2 in almost all kinds of tissues and cells, such as the brain, lung, gastrointestinal tract, testis, and some kind of cancer cells [19]. In an overall view, the function and mechanism of ERK1/2 vary under different circumstances and physiological statuses [20]. Similarly, some experiments have shown that the expression of ERK1 and ERK2 are widely presented in different parts of the reproductive organs in mammals and poultry [21,22,23,24,25,26,27,28,29,30,31,32]. However, the whole profile and specific role of ERK1/2 expression in reproductive organs, along with the different reproductive cycle stages, have not yet been detected and analyzed clearly in mammals.

The yak (*Bos grunniens*) is a kind of seasonal breeding-mammal that is mostly allocated around the Qinghai-Tibet Plateau [33]. Because of their predominant adaptability to high altitude, cold conditions, nutrition deficiency, and hypoxia environments, yaks have always been believed to be the most important means of production and livelihood of local herdsmen [34]. However, affected by formidable natural conditions, the yak presents a very low reproductive efficiency. The majority of them can give birth only once every two years or twice every three years [35,36]. Therefore, it is an important aspect—for improving the reproductive efficiency—to investigate the regulation of breeding activities [37,38,39]. Similar to other mammals, the female reproductive organs of adult yaks experience cyclic variation during different stages of their reproductive cycles. Ovary follicle development, corpus luteum generation, luteolysis, uterine distention, and placentation take place at specific periods [22,40,41]. Along with the reproductive cycle, numerous cells, such as ovarian granular cells, luteal cells, endometrial epithelium cells, and endometrial stromal cells experience proliferation or apoptosis [42]. As previously noted, ERK1/2 are involved in cell growth, differentiation, and apoptosis. All of these processes occur in the ovary and uterus during normal reproductive cycles. Therefore, it should be an attractive prospect to explore the roles and regulatory mechanisms of ERK1/2 in yak reproduction. We hypothesized that the abundance of ERKs would fluctuate in the reproductive tract during the different reproductive situations.

## 2. Materials and Methods

### 2.1. Samples of Yak’s Reproducitve Organs

All procedures involving animals were approved by the Animal Care and Use Committee of Gansu Agricultural University. Samples of ovaries, oviducts, and uterus were incised from yaks (5 yaks per group) within 10 min after being slaughtered, in Xining abattoir of Qinghai Province, China. The yaks used for sample collection were estimated to be between 5 and 8 years old, were inspected to have no clinical disease, and no obvious pathological changes. For mRNA and protein analysis of ERK1/2 expression using WB and RT-PCR, tissue samples were immersed into liquid nitrogen for storage immediately after washing with 0.1% diethylpyrocarbonate (DEPC) reated water. Another tissue sample of the same yak was cut into small pieces and fixed with 4% paraformaldehyde phosphate buffer (pH 7.3) in 4 °C, at least 2 weeks before subsequent use.

According to the status of the yak’s reproductive organ, we divided these tissue samples into 3 groups. The follicular phase group: there was only one ≥10.0 mm follicle and no macroscopically corpus luteum in both ovaries; the two uterine horns were symmetric and not dilated. The luteal phase group: there was only one ≥10.0 mm functional corpus luteum and no >8.0 mm follicle. Smaller luteum existed in both ovaries, and the two uterine horns were symmetric, not dilated. The gestation period group: functional corpus luteum was present in one of the two ovaries; one side of the uterine horns was obviously dilated and contained fetal.

### 2.2. erk1 and erk2 Gene Expression Analysis

Total RNA of yak tissue samples of every group were extracted using TRIzol reagent (Invitrogen, Carlsbad, CA, USA). Experion RNA StdSens Analysis Kit (BioRad, Munich, Germany) was used for total RNA quality and quantity assessment on Experion Automated Electrophoresis Station (BioRad, Munich, Germany). The value of RNA quality indicator (RQI) adopted was between 5 and 10. To avoid genomic DNA contamination, enzymatic digestion was performed with RNase-free DNase I (Omega, Norcross, GA, USA). Subsequently, with MOligo-dT18 primers, total RNA was reverse transcribed into cDNA using RevertAid first Strand cDNA Synthesis Kit (Promega, Mannheim, Germany). For relative quantitative analysis of gene expression, the *erk1* and *erk2* primers were designed according to bovine sequences (NM001110018.1, NM175793.2), and *β-actin* primers were designed based on the yak sequences (NM001034034.2). *erk1* (F: 5′-ATCCCTTGGCTGTCG-3′, R: 5′–AGGCGTTTCCATTCGT-3′). *erk2* (F: 5′-ATCCCTTGGCTGTCG-3′, R: 5′-AGGCGTTTCCATTCGT-3′). *β-actin* (F: 5′-AGGCTGTGCTGTCCCTGTATG-3′, R: 5′–GCTCGGCTGTGGTGGTAAA-3′). The predicted product length of these primers was 107, 111, and 187 bp, respectively. Real-time fluorescent quantitative PCR system (Light Cycler 480, Roche, Germany) was used to perform Real Time PCR analysis, as previously described [41]. Briefly, 200 ng of total cDNA was amplified in a 20 μL reaction mixture containing 10 μL SYBR Premix Dimer Eraser (Promega, Mannheim, Germany) and 100 nM of forward and reverse primer. The optimized 0conditions of RT-PCR were listed as below: pre-denaturation at 95 °C for 4 min, denaturing 40 cycles at 95 °C for 30 s, annealing at 58 °C for 30 s, and extension fluorescence acquisition at 72 °C for 25 s. The specificity analysis was performed through melting curve from 65 °C to 95 °C in 0.5 °C steps, each lasting 5 s, and the product was conduct electrophoresis in 2.5% agarose gel to confirm correct size.

### 2.3. The Quantitative Analysis of ERK1/2 Proteins Expression

For quantitative analysis of ERK1 and ERK2 proteins expression, frozen tissues were taken out of liquid nitrogen. After, they were washed with cold phosphate buffer saline (PBS) for three times. The total proteins of samples were extracted using Beyotime extraction buffer, and the protein concentration was detected via Bradford assay kit (Bio-Rad, Hercules, CA, USA). Then, the protein suspension was conducted, electrophoresis (PAGE), filled on 10% SDS-PAGE gel. Subsequently, separated proteins were electrophoretically transferred onto enhanced chemiluminescence (ECL) polyvinylidene fluoride (PVDF) membranes (Amersham, Piscataway, NJ, USA), using a mini transfer instrument for electrophoresis (Bio-Rad, Hercules, CA, USA) at 300 mA for 90 min. After, it was blocked with Tris-buffered saline (contained 5% non-fat dry milk and 0.1% Tween-20) the protein was incubated with a primary antibody against ERK1/2 (p44/42 MAPKRabbit mAb, 4695 s, CST, Danvers, MA, USA) and β-actin (ß-Actin Rabbit mAb, 4970 s, CST, Danvers, MA, USA) in Tris-buffered saline at 37 °C for 2 h. After, it was washed in Tris-buffered saline, the protein reacted with a secondary antibody (goat anti-rabbit immunoglobulin G conjugated with horseradish peroxidase, sc-2030, Santa Cruz Biotechnology, Santa Cruz, CA, USA) under 37 °C for another 2 h. After it was washed with Tris-buffered saline, the membrane reacted with ECL Substrate kit (ab65623, Abcam, Cambridge, UK). For negative controls (were conducted using normal IgG reagent) replaced the primary antibodies. The protein band’s intensity on the membranes was measured using the densitometric analysis system (Bio-Rad, Hercules, CA, USA). Relative intensity of ERK1/2 proteins were normalized with β-actin bands.

### 2.4. Localizational Analysis of ERK1/2 Proteins Expression

The localization of ERK1/2 proteins was analyzed on paraffin-embedded 4 μm tissue sections using the immunohistochemical technique. At first, paraffin-embedded tissue sections were dewaxed and rehydrated using gradient acetone and alcohol solution, and eliminated endogenous peroxidase activity by incubating with 3% H_2_O_2_ for 5 min. Then, the tissue sections were immersed in citrate buffer (pH 6.0) and boiled for 10 min to enhance the antigen activity. Subsequently, for reducing non-specific binding of the primary antibody, sections were incubated with normal goat blocking serum for 15 min after being thoroughly washed with PBS. After that, tissue sections were incubated with primary antibody (1:450 diluted p44/42 MAPK Rabbit mAb, 4695 s, Cell Signaling Technology, Inc., Danvers, MA, USA) at 37 °C for 120 min, followed by incubation with secondary antibody (biotinylated goat anti-rabbit IgG, Invitrogen Zymed Laboratories, Carlsbad, CA, USA) for 15 min at 37 °C. Then, sections were washed another three times in PBS and reacted with avidin-biotin peroxidase. The immunoperoxidase color reaction was completed by adding 3,3′-diaminobenzidine agent (DAB, Invitrogen Zymed Laboratories, Carlsbad, CA, USA) as substrate. After a moderate brown reaction product presented, tissue sections were timely washed with PBS to stop color reaction. In the end, after being counterstained with haematoxylin, tissue sections were mounted with resin, for observation under microscope and storage. We also performed the negative controls experiment by replacing the primary antibody with normal non-immune IgG agent.

### 2.5. Data Analysis

The relative quantity of *erk1* and *erk2* mRNA and protein expression are presented as fold increase, where expression of target mRNA or proteins was divided by the expression of ß-Actin. The difference of expression quantity between reproductive stages was analyzed by one-way ANOVA procedure using SPSS 19.0 (SPSS Inc., Chicago, USA). It was considered to be statistically significant at a probability of *p* < 0.05; meanwhile, extremely significant was defined as *p* < 0.01.

## 3. Results

### 3.1. mRNA Expression of erk1 and erk2 in Female Yak Reproductive Organs

A variation of *erk1* and *erk2* gene expression was detected in female yak reproduction organs during different stages of the reproductive cycle. In the ovary, the *erk1* gene expression was significantly higher in the luteal phase (*p* < 0.05) and gestation period (*p* < 0.01) than the follicular phase. Similar to *erk1*, the mRNA level of *erk2* was also higher in the luteal phase compared with the follicular phase and gestation period (*p* < 0.05), but no significant difference existed between gestation period and follicular phase (*p* > 0.05). In the oviduct, the level of the *erk1* gene expression was highest in the luteal phase, which was significantly different with that in the follicular phase and gestation period (*p* < 0.05). However, the *erk2* gene expression was extremely higher in the follicular phase than in the luteal phase (*p* < 0.05) and gestation period (*p* < 0.01). In the uterus, the *erk1* gene expression was extremely higher in the luteal phase and gestation period than in the follicular phase (*p* < 0.05), although no significant difference was found between the luteal phase and gestation period. Similarly, the level of *erk2* gene expression was also higher in the luteal phase and gestation period than that in the follicular phase (*p* < 0.05) (Figure 1).

### 3.2. ERK1 and ERK2 Protein Expression in the Female Yak’s Reproductive Organs

Both ERK1 and ERK2 proteins were detected using the western blotting technique. The characteristics of ERK1 and ERK2 protein expression in the female yak’s reproductive organs at different reproductive statuses were similar as their mRNA expression. In ovaries, the level of ERK1 protein expression was nearly twofold of β-actin during the gestation period, while it only approximated one-half of β-actin in the follicular phase. The difference between these two stages was extremely significant (*p* < 0.01). The expression of ERK2 was higher in the luteal phase than that in the follicular phase and gestation period (*p* < 0.05). A little higher ERK2 expression was also found in the gestation period compared with the follicular phase, although there was no significant difference between these two groups (*p* > 0.05). In the oviduct, the level of ERK1 protein expression was highest in the luteal phase, which was significantly different with that in the follicular phase and gestation period (*p* < 0.01). However, ERK2 protein expression was extremely higher, in both the follicular phase and luteal phase, than that in the gestation period (*p* < 0.01). In the uterus, both ERK1 and ERK2 protein expressions were extremely higher in the luteal phase and gestation period, compared with the follicular phase (*p* < 0.01), although the difference between the luteal phase and gestation period was not significant (Figure 2).

### 3.3. Immunolocalization of ERK1/2 Proteins in the Female Yak’s Reproductive Organs

Immunohistochemical analysis revealed a light to dense positive reaction accumulation of signals for ERK1/2 proteins in the female yak’s reproductive organs under different reproductive statuses. In the ovary, ERK1/2 proteins were mainly expressed in surface epithelium, follicular granulosa cells, ovarian stroma, vascular endothelium, and corpus luteum. In particular, in the corpus luteum of the gestation period, most granular luteal cells were stained into strong brown, while membranous luteal cells were not stained. However, the overall signal intensity of ERK1/2 proteins in the ovary of the luteal phase was less pronounced compared with that of the follicular phase and gestation period (Figure 3A–C). In the oviduct, the positive accumulations of signals for ERK1/2 proteins localized moderately in villous epithelial cells, and some dispersed villous stromal cells. The intensity of ERK1/2 proteins expression was slightly higher in the oviduct of the luteal phase and the follicular phase compared with that of the gestation period (Figure 3E–G). In the uterus, the main compartments of ERK1/2 protein expressions were endometrial luminal epithelium, glandular epithelium, endometrial stroma, and vascular endothelium. In addition, ERK1/2 protein signals also moderately appeared in the myometrium of the luteal phase and gestation period. The signal intensity of ERK1/2 proteins in the uterus of luteal phase and gestation period was obviously higher than that of the follicular phase (Figure 3I–K).

## 4. Discussion

The present study described the expression of ERK1 and ERK2 proteins and mRNA in female reproductive organs of the adult yak during different the stage of the reproductive cycle based on RT-PCR, western blot, and immunohistochemistry analysis. The results showed that both the intensity and distribution of ERK1 and ERK2 expression in the ovary, oviduct, and uterus varied with the stage of the reproductive cycle. In general, the expression of ERK1 and ERK2 proteins and mRNA was most pronounced in the ovary of the luteal phase and gestation period, the oviduct of the luteal phase and the uterus of the gestation period. The histological appearance and physiological process of the main reproductive organs also varies with the different reproductive stages [43].

In the ovary, the results of the present study indicated that both the ERK1 and ERK2 proteins and mRNA were highly expressed during the luteal phase and gestation period, and the immunohistochemical analysis revealed an intense distribution of the ERK1/2 protein in follicular granulosa cells and corpus luteum cells. It is well know that the ovary is a highly organized composite of oocytes, granulosa cells, stromal cells, and sometimes corpus luteum cells whose interaction induce development of follicles, ovulation, formation of corpus luteum, and luteolysis during different stages of the reproductive cycle; many factors involved in the regulation mechanism of the ovary’s physiological process. The classical point of view believes that the development of the ovary is mainly regulated by the hypothalamic-pituitary-ovarian axis. Therefore, various reproductive hormones play a very important role as external signals. Recently, studies have focused on the role and pathway of intra-ovarian signaling cascades in regulating ovary development [44]. As we discuss in the present study, ERK1/2 is one of the important intra-ovarian regulators of ovary development in mammals. The ERK1/2 pathway is intensively studied in follicle development of mice and other mammal species [45,46,47,48]. It has been reported that ERK1/2, and some other protein kinases, impact the cumulus expansion and oocyte maturation of porcine cumulus oocyte complexes (COCs) by inducing the expression of both the epidermal growth factor (EGF)-like factor and its protease [49]. Using cultured primary rat granulosa, Wayne et al. proved that follicle-stimulating hormone (FSH) and luteinizing hormone (LH) control granulosa cell function and differentiation by activating Ras protein, and some downstream kinases, especially MEK1 and ERK1/2 [50]. On the contrary, the experiment conducted on gene knockout mice and pharmacologically inhibited materials also demonstrated a pivotal role of the ERK1/2 pathway in ovulatory processes [51,52,53]. Most recently, by injection of pharmacological inhibitor (PD0325901) of ERK1/2 into the pre-ovulatory dominant follicle, Yasmin et al. found that four of five cows failed to ovulate, and 285 differentially expressed genes were identified from granulosa cells of drug treated follicles. Based on the analysis of the differentially expressed genes, they further affirmed a significant role of ERK1/2 in mediating LH induced gene expression in ovulating follicles, and the physiological process of ovulation dependent on proper ERK1/2 signaling in bovine [25]. Ryan et al. also found that the level of ERK signal proteins was different between dominant and subordinate follicles early in the stage of dominant follicle selection in cows [26]. However, there is no report about the expression of ERK1/2 in the ovary of the yak. The mass emergence of the ERK1/2 protein in follicular granulosa cells, demonstrated by the present study, probably means that the follicular development and further ovulation is a physiological process dependent on the regulation of ERK1/2.

Our study also demonstrates an intense expression of ERK1/2 in corpus luteum of the luteal phase and gestation period. It is well known that corpus luteum is a transient endocrine tissue that is derived from a pre-ovulatory follicle (Graffian follicle). It plays a significant role in controlling the reproductive cycle of mammals. The degeneration of corpus luteum, in case of un-pregnancy, is a key event for the initiation of a new reproductive cycle, re-ovulation, and obtaining the next chance of conception. On the contrary, the prolongation of the luteal function is also essential for the development of the embryo and maintenance of pregnancy relationship [54]. Generally, prostaglandin F_2α_ (PGF_2α_), as an extra-ovarian physiological luteolysin derived from uterus, is the key factor inducing the degeneration of the corpus luteum. It is well recognized that apoptosis is the main mechanism of PGF_2α_ induced regression of corpus luteum in bovine and other mammals [54]. As an important regulatory factor of apoptosis and proliferation of many types of cells, ERK1/2 is also involved in the mechanism of regression and functional maintenance of corpus luteum. For example, Maekawa et al. have found that human chorionic gonadotrophin (hCG) increased the StAR gene (coding histone modification enzymes) expression in granulosa cells through the ERK1/2 mediated signal pathway in the physiological process of follicular luteinization [55]. This is one of a few reports about the role of ERK1/2 in luteal formation. Strikingly, numerous research indicates that ERK1/2 is related to the regression of corpus luteum. Based on the experiment performed on Sprague-Dawley rats, Choi et al. found that PGF2α inducted luteal cell autophagy was accompanied with the activation of ERK1/2 during corpus luteum regression, and it is not regulated by the mammalian target of rapamycin (mTOR) signal [27]. Qi L et al. also proved that prostaglandin F (PGF) treatment increased ERK1/2 and signal transducers and activators of transcription 3 (STAT3) phosphorylation, and this is a probable molecular mechanism of luteal regression in pseudopregnant rats [56]. A similar conclusion was obtained by Chen, who demonstrated that ERK1/2 signaling cascade can be activated by other molecular mechanisms in bovine luteal cells [57].

In addition, many different stimuli can induce the change of intracellular signal transduction. To analyze the change of intracellular mitogen-activated protein kinase (MAPK) signaling cascade induced by stress-related signaling events. Rueda et al. detected the phosphorylation level of main MAPK members in cultured luteal cells. They found that both jun-n-terminal kinase (JNK) and p38MAPK were highly phosphorylated after UV treatment, but the phosphorylation of ERK1 and ERK2 was low. In addition, all of these changes were related to a high apoptosis rate of in vitro cultured luteal cells. Based on these experimental results, they believed that stress signals induced regression of corpus luteum, perhaps mediated by the activation of MAPK cascade [58]. As we have mentioned above, the yak is a kind of livestock living in a very harsh environment. In particular, the gestation period of the female yak is mainly spent in winter and spring, when low temperature, hypoxia, and nutrition deficiency threaten them at all times. The high expression of ERK1/2 in corpus luteum perhaps is a mechanism to maintain the luteal function and pregnancy; thus, yaks developed a prominent adaptability to the rigorous natural environment.

In the oviduct, our present study has also found a more intense expression of both ERK1 protein and mRNA in the luteal phase compared with the follicular phase and gestation period of the female yak reproductive cycle. Further immunolocalization analysis indicated that ERK1/2 proteins mainly localized in villous epithelial cells. As we all know, the mammalian oviduct is a convoluted tube bridging the ovary and the uterus in the reproductive process of mammals. It also provides a suitable place for fertilization and development of preimplantation embryos. The function of the ovary and uterus in reproductive activities is always an important issue attracting great attention, and has widely been investigated for a long times. However, it seems that we neglect the importance of the oviduct in reproduction, and relative information about the oviduct is insufficiently. Recently, some researchers begin to focus on the important role of the oviduct, both in natural fertilization and in pre-implantation embryo development. The correct biophysical function, such as a moderate and rhythmic smooth muscle contraction, ensures the transportation of the gametes and zygote in the oviduct. Meanwhile, a suitable biochemical component of the oviduct content provides an adapt circumstance for the development of the preimplantation embryo. On the contrary, the incorrect function of the oviduct may be the result in infertility, or deficiency of the embryonal development [59,60]. The oviduct content, also named as oviduct fluid, consists of proteins secreted by the secretory cells of the oviduct epithelium and other plasma-derived constituents [61,62]. Therefore, a normal function of this epithelium and its secretions is a basic requirement for successful fertilization and establishment of pregnancy. Recently, numerous genes and proteins were identified involving in the regulation of oviduct functions, using transcriptomic and proteomic techniques [63]. For instance, Cerny et al. performed a transcriptome analysis of bovine oviduct epithelial cells, and they found that a large number of genes differentially expressed during different estrus stages [64]. In line with these observations, significant differences of the profile of the gene and protein expression in the oviduct, between different estrus cycle stages, have also been reported in ewes [65], bovine [66], pigs [67,68], and humans [69,70]. For example, in spontaneous estrous ewes, a total of 280 proteins were identified in the oviduct, and 64 proteins upregulated during estrus, while 17 proteins were upregulated in the luteal phase [65]. However, there is no report about the expression of ERK1 and ERK2 in the oviduct of the mammalian species.

The result obtained from the oviduct of different stages of the yak’s reproductive cycle in the present study suggests that, as an intracellular signal transduction pathway, ERK1/2 plays a significant role in regulating the secretory activity of the oviduct epithelium, and provides an optimal oviduct microenvironment for oocyte fertilization and embryo development before implantation.

In the uterus, we also observed a more strong expression of ERK1 and ERK2 during the luteal phase and gestation period. The positive reaction of the ERK1/2 proteins were located strongly in the endometrial epithelium cells, stroma cells, and moderately in the uterine smooth muscle cells. In the follicular phase, under the stimulation of high concentrations of FSH and LH, follicular recruitment and abundant E_2_ secretion take place in the ovary, but the endometrium thickness and P_4_ level are minimal. In the same times, numerous of endometrial epithelium cells undergo degeneration and necrosis [71,72]. After ovulation (luteal phase), a new corpus luteum begins to develop and produces a large amount of progesterone. Then, the differentiation and proliferation of the endometrial epithelium cells take place under the stimulation of progesterone, to make preparation for embryo implantation [71,72].

The essential role of P_4_ and E_2_ in regulating endometrial differentiation, growth, and receptivity to implantation during primate menstrual cycle has been widely recognized [73,74,75,76]. Numbers of reports also demonstrated that MAPK signaling pathways are activated in the uterus during the process of embryos implantation in the rat [75,77] and human [78]. Sayem’s experiment indicated that in female rats, thyroxin treatment increased the expression of ERK1/2 proteins in uterine stroma, which could help the uterine to adopt the implantation of embryos [79]. However, most of the materials are obtained from the estrus cycle or early gestation period (before embryo implantation). A few researchers also pay attention to the role of ERK in the uterus during the whole gestation period. For example, Welshet al. proved that estrogenic actions were mediated by the estrogen receptor α via activating ERK signals in human myometrium during pregnancy [80].

According to the analysis above, we consider that the expression of ERK1 and ERK2 in yak’s endometrial glandular and luminal epithelial cells and uterine smooth muscle cells during the luteal phase and gestation period may play an important role in glandular secretion, and maintain an appropriate muscle tension of the uterus. This is essential for embryo implantation and subsequent development.

## 5. Conclusions

The present study firstly demonstrates the wide expression of ERK1 and ERK2 proteins and mRNA in female reproductive organs of the adult yak. The intensity of ERK1 and ERK2 proteins and their mRNA expression in the yak’s ovary, oviduct, and uterus varies with the stage of the reproductive cycle. The variation character of ERK1 and ERK 2 expression in the yak’s main reproductive organs during different stages implies that they play an important role in the regulation of reproductive functions under different physiological situations.

## Figures and Tables

**Figure 1 animals-10-00334-f001:**
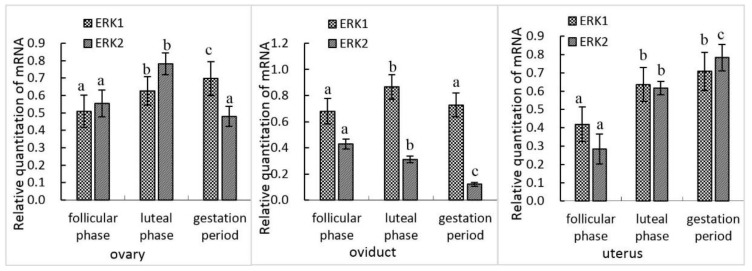
Expression of extracellular signal-regulated kinase (*ERK)1* and *ERK*2 mRNA in female reproductive organs at different stages. In a certain reproductive organ, the difference of the same protein between a and b or b and c is significant (0.05 > *p* > 0.01), and the difference of the same protein between a and c is extremely significant (*p* < 0.01).

**Figure 2 animals-10-00334-f002:**
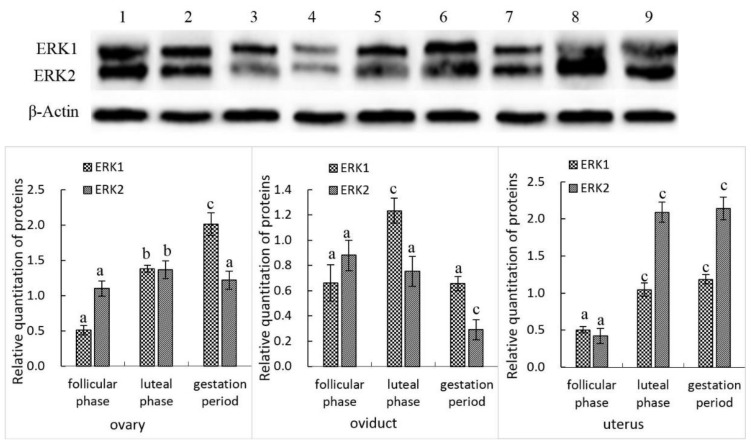
The western blot bands and expression analysis of ERK1, ERK2, and β-Actin proteins; 1, 2, 3: ovary; 4, 5, 6: oviduct; 7, 8, 9: uterus; 1, 4, 7: follicular phase; 2, 5, 8: luteal phase; 3, 6, 9: gestation period. Note: In certain reproductive organs, the difference of the same protein between a and b or b and c is significant (0.05 > *p* > 0.01), and the difference of the same protein between a and c is extremely significant (*p* < 0.01).

**Figure 3 animals-10-00334-f003:**
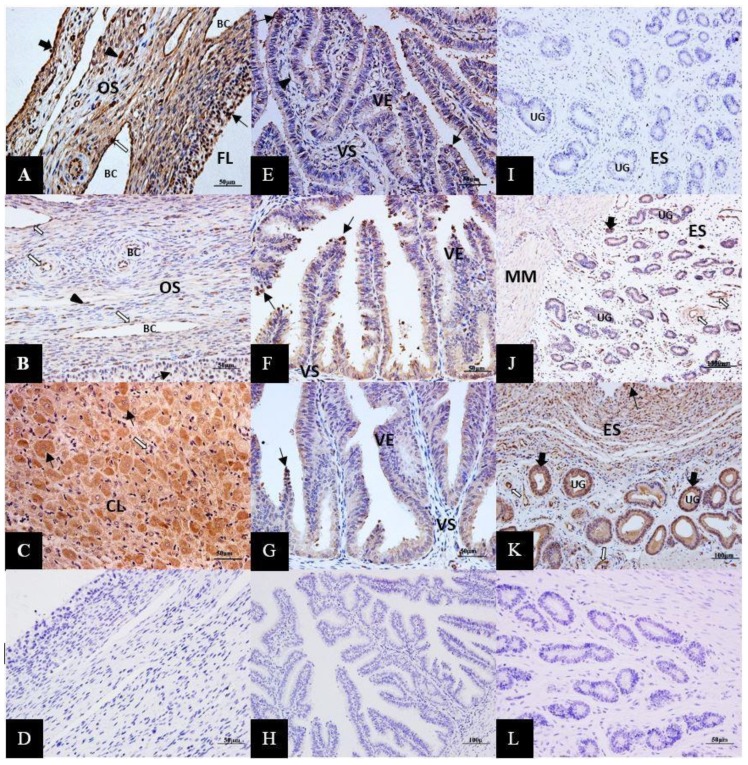
Immunolocalization of ERK1/2 proteins (brown stained regions) in the female yak’s ovary (**A**,**B**,**C**), oviduct (**E**–**G**), and uterus (**I**,**J**,**K**) during the follicular phase (**A**,**E**,**I**), luteal phase (**B**,**F**,**J**), and gestation period (**C**,**G**,**K**). (**A**) ERK1/2 appeared in follicular granulosa cells (thin arrow), ovarian stroma cells (arrowhead), vascular endothelium (thick arrow), and surface epithelium (white thick arrow) of the ovary during the follicular phase. (**B**) ERK1/2 proteins moderately expressed in vascular endothelium (thick arrow), vascular endothelium (thick arrow), and follicular granulosa cells (thin arrow) of the ovary during the luteal phase. (**C**) Most of granular luteal cells (thin arrow) were stained into strong brown, while membranous luteal cells (white thick arrow) in corpus luteum of gestation period. (**D**) Negative control, ovary tissue sections were incubated with an equivalent non-immune IgG agent instead of a rabbit polyclonal antibody to ERK1/2. (**E**,**F**,**G**). The positive reaction of ERK1/2 proteins moderately localized in villous epithelial cells (thin arrow) and some dispersed villous stromal cells (arrowhead) of the oviduct. The intensity of ERK1/2 proteins expression was slight higher in the luteal phase (**F**) and follicular phase (**E**) compared with that of the gestation period (**F**). (**H**) Negative control of oviduct sections. (**I**,**J**,**K**) The main compartments of the ERK1/2 protein expression were endometrial epithelium (thin arrow), glandular epithelium (thick arrow), endometrial stroma cells (arrowhead), and vascular endothelium (white thick arrow) of the uterus. The signal intensity of ERK1/2 proteins in the uterus of the luteal phase (**J**) and gestation period (**K**) was obviously higher than that of the follicular phase (**I**). (**L**) Negative control of the uterus section. FL: follicle, BC: blood capillary, OS: ovarian stroma, CL: corpus luteum, VE: villous epithelium, VS: villous stroma, UG: uterine gland, ES: endometrial stroma, MM: myometrium.

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
