# Peer review of "The Expression of ERK1/2 in Female Yak (Bos grunniens) Reproductive Organs"

_animals, 2020, doi:10.3390/ani10020334_

Round 1

Reviewer 1 Report

General comments:

The aim of the present study was to evaluate the expression and distribution of ERK1/2 in the reproductive tract at different stages of females yaks. Overall, the manuscript is well structured and written without major problems. Nevertheless, some points reported in the specific comments should be addressed. Special attention needs to be given to characterize the 3 groups of females better. I also suggest don’t consider, throughout the text, the gestation period as a phase of the oestrus cycle (there are only two classical phases as you, and the gestation as a physiologic stage).

L40:  “…MAPK…” instead of “…Mitogen-activated protein kinases (MAPK)…”

L233-236: Why these two sentences were inserted where? The stage of the reproductive cycle was established only by a single morphological evaluation. These can originate some criticisms:  Follicular phase: there is a chance of anestrus for some female, even a 10 mm follicle is present; also note if there are two successive cycles, the corpus luteum is in regression and well detected in the ovary? Luteal phase: there is a chance of early gestation for some female, previously to the dilation of the pregnant uterine horn? Also here the size of corpus luteum is not definitive evidence that the female is not pregnant. If possible, I suggest improving the M&M (L93-99) at this level.

L236-237: Also here, this sentence reports the main aim of the present study and should be removed or moved to the Introduction section.

L272: The human is a mammal.

L289-291: There are recent studies published in cows involving regulation of ERK1/2 in the ovulation process. I suggest citing at least one.

L299- The function of the PGF2α is luteolytic. The word “believed” should be removed.

Author Response

Response to Reviewer 1 Comments

Point 1: The aim of the present study was to evaluate the expression and distribution of ERK1/2 in the reproductive tract at different stages of females yaks. Overall, the manuscript is well structured and written without major problems. Nevertheless, some points reported in the specific comments should be addressed. Special attention needs to be given to characterize the 3 groups of females better. I also suggest don’t consider, throughout the text, the gestation period as a phase of the oestrus cycle (there are only two classical phases as you, and the gestation as a physiologic stage)

Response 1: Many thanks for your positive comments to our research works. We have carefully studied the issues according to your suggestion, and have revised the manuscript in the revision. We used the phrase of “gestation period” to describe a stage of reproductive cycle, but not oestrus cycle throughout the text. As you pointed out, oestrus cycle can be devided into two phase: follicular phase and Luteal phase. But  the reproductive cycle usually include more stages than oestrus cycle, such as gestation period, Postpartum period and lactation period. In view of the effection of these stage to reproductive activity and the convenience of tissue sample collecting and grouping, we selected these three stages in our research.

Point 2: L40:  “…MAPK…” instead of “…Mitogen-activated protein kinases (MAPK)…”

Response 2: It has been revised.

Point 3: L233-236: Why these two sentences were inserted where? The stage of the reproductive cycle was established only by a single morphological evaluation. These can originate some criticisms:  Follicular phase: there is a chance of anestrus for some female, even a 10 mm follicle is present; also note if there are two successive cycles, the corpus luteum is in regression and well detected in the ovary? Luteal phase: there is a chance of early gestation for some female, previously to the dilation of the pregnant uterine horn? Also here the size of corpus luteum is not definitive evidence that the female is not pregnant. If possible, I suggest improving the M&M (L93-99) at this level.

Response 3: These two sentences has been revised. We have spend a long time to think the problem of how to divide these samples into different groups, and what names would be more suitable for these different groups. According to the classical theriogenology,  follicular phase and luteum phase are divided by the different histomorphological characteristic of ovary. Follicular phase not means estrus which was defined  mainly based on the animals behaviour. The difference of the ovary between follicular phase and luteum phase is distinguishable, and our samples of these two stages were divided undoubtly. Just as you have mentioned, The difficulty lies in the distinguishment of luteum phase and early gestation period. In fact, the luteum phase partly overlaps  with early gestation period. Although we unable to exclude the chance of pregnant, the samples divided into luteum phase was collected from the yaks of no obvious symptoms of pregnancy, and these samples were obviously different from that of gestation period in our experiment. We would like to give our sincere thanks to you if you can  provide us some specific guidances about the grouping of the samples.   

Point 4: L236-237: Also here, this sentence reports the main aim of the present study and should be removed or moved to the Introduction section.

Response 4: This sentence has been moved to the Introduction section.

Point 5: L272: The human is a mammal.

Response 5: It has been revised.

Point 6:L289-291: There are recent studies published in cows involving regulation of ERK1/2 in the ovulation process. I suggest citing at least one.

Response 6: In fact, we have cited one article relating to the ERK1/2 and ovulation process (reference 25). In the revision, we cited another one (reference 26).

Point 7:L299- The function of the PGF2α is luteolytic. The word “believed” should be removed.

Response 7: It has been revised.

Reviewer 2 Report

Potentially good and unique paper with expression of ERK1/2 in female yak. Here are some suggestions to improve this paper:

One of the major concerns is use of bovine sequences for the expression study of erk1 and erk2 (line 111). Why yak specific sequence is not used in this analysis? Does is affect the interpretation of results? However, beta actin (controls) were yak specific (line 112). It is little confusing, could you elaborate/explain about this.

In the same line of assessment, is ERK gene conserved across similar species such as yak, cattle, sheep, and others. Can you mention sequence similarity or dissimilarity of this gene across species in introduction section?

Study population needs to be more detailed. Are those adult female yaks from similar age or herds or it is completely random?

Are 5 animals per group is enough to detect the differences in expression? What is the power of this analysis/study?

Author Response

Response to Reviewer 2 Comments

Point 1: One of the major concerns is use of bovine sequences for the expression study of erk1 and erk2 (line 111). Why yak specific sequence is not used in this analysis? Does is affect the interpretation of results? However, beta actin (controls) were yak specific (line 112). It is little confusing, could you elaborate/explain about this.

Response 1: There was no submitted yak’s sequence information of erk1 and erk2 on GenBank database at the time of our experiment performed, while beta actin gene was available. So we cloned and analysed these two genes of yak, and submitted the sequences infornation to the GenBank for the first time (the serial number are MK387870 and  MK784020, respectively). Therefore, we designed the primer of erk1 and erk2 referenced the nearest species, the bovine. In view of the amplified product, it was in according with the predicted length of these primers. So we consider that the results are reliable.

Point 2: In the same line of assessment, is ERK gene conserved across similar species such as yak, cattle, sheep, and others. Can you mention sequence similarity or dissimilarity of this gene across species in introduction section?

Response 2: According to the previous argument,  ERK gene family is evolutionnarily conserved and is found in all eukaryotes, including yeasts, plants, vertebrates and invertebrates (Widmann C, Gibson S, Jarpe MB and Johnson GL. Mitogen-activated protein kinase: conservation of a three-kinase module from yeast to human. Physiol Rev. 1999; 79(1): 143–180). In view of our sequencing results, the sequence similarity of both erk1 and erk2 are above 98%  between yak (Bos grunniens) and Bos taurus. The gene clone and bioinformatics analysis of these two gene was included in another manuscript which would published in recently. We have added a sentence in introduction section to state the issue about ERK gene conservation.

Point 3: Study population needs to be more detailed. Are those adult female yaks from similar age or herds or it is completely random?

Response 3: The yaks used for samples colletion were selected between 5~8 years old, but not come from a same herd. The information about the yak’s age and other selection criteria has been added in the materials and methods section of revision.

Point 4: Are 5 animals per group is enough to detect the differences in expression? What is the power of this analysis/study?

Response 4: We have referred some articles performing the detection of gene and protein expression in domestic animals, the number of samples also no more than 5. We also have recognized that 5 yaks are not enough to form a perfect small sample from a statistical perspective. However, in view of our experiment results, we found that the original data of RT-PCT and WB are closely between different individual of the same group. Therefore, we believed that our results can represent the overall situation of every group to some degree.